# BRENT: Bidirectional Retrieval Enhanced Norwegian Transformer

**Lucas Georges Gabriel Charpentier,\* Sondre Wold,\***
**David Samuel and Egil Rønningstad**

University of Oslo, Language Technology Group
{lgcharpe|sondrewo|egilron|davisamu}@ifi.uio.no

## Abstract

Retrieval-based language models are increasingly employed in question-answering tasks. These models search in a corpus of documents for relevant information instead of having all factual knowledge stored in its parameters, thereby enhancing efficiency, transparency, and adaptability. We develop the first Norwegian retrieval-based model by adapting the REALM framework and evaluate it on various tasks. After training, we also separate the language model, which we call the *reader*, from the retriever components, and show that this can be fine-tuned on a range of downstream tasks. Results show that retrieval augmented language modeling improves the reader's performance on extractive question-answering, suggesting that this type of training improves language models' general ability to use context and that this does not happen at the expense of other abilities such as part-of-speech tagging, dependency parsing, named entity recognition, and lemmatization. Code, trained models, and data are made publicly available.[1]

## 1 Introduction

Retrieval-based language models meet some important shortcomings associated with pre-trained language models (PLMs): they are more dynamic, allowing for updating of knowledge without having to re-train the model from scratch; they are more transparent, allowing backtracking the source of returned statements; and they are more efficient, as retrieval provides a non-parametric memory. The accentuated benefit of these models has been the

---

\*The authors contributed equally to this work
[1]https://github.com/ltgoslo/brent

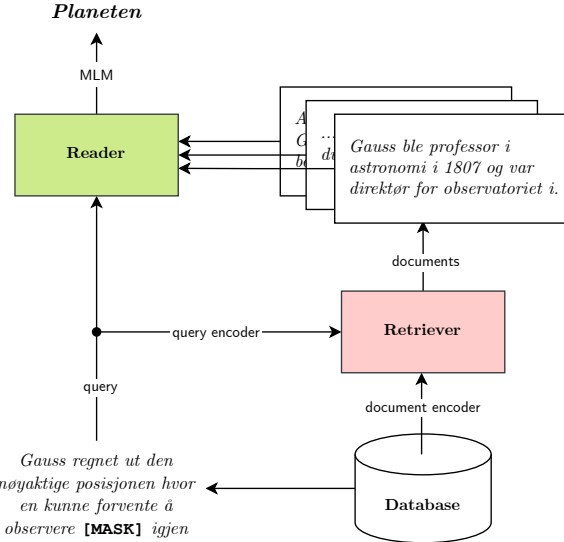

Figure 1: The proposed architecture, based on the REALM method from Guu et al. (2020).

OpenQA task – where they have established new state-of-the-art results on datasets like NaturalQuestions (Kwiatkowski et al., 2019) and WebQuestions (Berant et al., 2013). There, models first fetch a relevant passage from a data source in order to be able to answer a question — as compared to extractive QA, where a passage with the correct answer is provided explicitly as additional input to the model, also referred to as machine reading comprehension.

In this work, we develop the first Norwegian retrieval-based model, BRENT: **B**idirectional **R**etrieval **E**nhanced **N**orwegian **T**ransformer, based on the general approach proposed by Guu et al. (2020). Our model consists of two *encoders* that respectively learn to embed documents and queries into dense vector representations, and a *reader* module that learns to utilize the retrieved context for prediction, as shown in Figure 1. These are trained jointly and end-to-end, and we start their training from an already pre-trained Norwegian LM. Compared to previous work, we use a

relatively small retrieval corpus consisting of 730k Wikipedia documents.

The learning objective is masked language modeling (MLM), and the top $k$ most relevant documents are retrieved from the retrieval corpus through a maximum inner product search (MIPS).

The size of our retrieval corpus allows us to update the search index synchronously during training and do exact matching, as opposed to the asynchronous updates and approximations done in Guu et al. (2020). Furthermore, we do not consider OpenQA as an evaluation task, but instead, we study how retrieval-augmented language modeling can be used as a continued pre-training step in order to improve context utilization in the reader model. That is, we evaluate the reader as a stand-alone — extracting it from the overall pipeline so that it can be distributed and used as a normal LM.

In order to analyze the effect of this continued pre-training, we also benchmark the reader against other NLP tasks that by intuition should not benefit from this type of training, such as part-of-speech-tagging, named entity recognition, dependency parsing, and lemmatization. We find that the retrieval-augmented training procedure increases the reader's performance on extractive QA without decreasing performance on these tasks. However, we find that it decreases performance on both targeted and sentence-level sentiment analysis. To summarize, our contributions are:

- We develop and release the first Norwegian retrieval-based language model.

- We study how retrieval improves the reader's ability to use context on extractive QA while still performing on par with comparable baselines on morpho-syntactic tasks.

- We analyze the different components of a retrieval-based system through a series of ablations, addressing problems associated with common design choices.

## 2 Related work

The basic setup for most retrieval-based approaches to NLP is that for a query $q$, be it a question for QA or a premise in natural language inference, the model must retrieve a set of passages relevant to $q$. Relevant candidates are then typically appended to $q$ before being passed to a classification layer.

While earlier work approached this using heuristics and sparse retrieval methods like BM25 (Robertson et al., 2009), recent work has focused on learning this retrieval step. Most of these use an architecture with an *encoder* and a *reader*: the encoder learns to represent $q$ and the retrieved passages in a representation space that makes it possible to match documents using the inner product operation, while the reader learns how to utilize the retrieved passage for downstream prediction. Recent work by Jiang et al. (2022) shows how it is also possible to model this interaction using a single transformer model (Vaswani et al., 2017) with a *retrieval as attention* technique, as compared to having separate encoders and readers.

Lee et al. (2019) note that it is computationally impractical to learn to make predictions conditioned on a retrieval corpus from scratch and thus proposed to pre-train the encoders with an inverse cloze task (ICT) in order to "prime" the model for retrieval. This is also done in Sachan et al. (2021). We outline more details on this and how we use ICT in the following section.

The most direct application of retrieval is to use a supervision signal such as OpenQA to train the context and passage encoders, such as in Khattab et al. (2021). However, Guu et al. (2020) show how this setup can also be used for language modeling. Using English Wikipedia as the retrieval corpus, they perform MLM conditioned on retrieved passages. Passages are retrieved using MIPS over an index that is asynchronously updated during training. They also use a masking technique that prioritizes named entities in order to incentivize the usage of world knowledge from the retrieved passages. A similar approach to language modeling is also done in Borgeaud et al. (2022), but over a corpus consisting of trillions of tokens. For both works, the LMs are trained for a number of steps with retrieval before being fine-tuned on a downstream task, as is the typical workflow with PLMs.

Lewis et al. (2020b) demonstrate how the encoder-reader architecture can be used for language generation as well. They propose both a sequence model where the generation is conditioned on the same set of retrieved documents for the entire sequence and a token model where a different document is used per target token. The retriever is based on Dense Passage Retrieval (DPR) (Karpukhin et al., 2020), which uses the same general approach to retrieval as Guu et al. (2020), where a PLM like BERT (Devlin et al., 2019) is

used as the encoder. The reader model is swapped with a generator based on BART (Lewis et al., 2020a).

# 3 Method

As in Guu et al. (2020), the architecture of BRENT can be separated into two parts: a retriever and a reader. Our architecture is modified to improve the speed of training, to ensure that the retrieved documents affect the predictions, and to incentivize the retrieval of world knowledge from the retrieval corpus instead of the reader memorizing it. This section puts forth the architecture and these modifications.

## 3.1 Architecture

**Retriever** The first part of BRENT is the retriever, which consists of two components: the Query Encoder ($\text{Enc}_{\text{query}}$) and the Document Encoder ($\text{Enc}_{\text{doc}}$). Both have their own sets of weights and in our case have a BERT-style architecture and tokenizer. However, these can be initialized with other types of dense representation learners and could potentially also share weights for faster training.

The retriever receives as input the query, $q$, which is the masked sentence from the pre-training corpus, and passes it to $\text{Enc}_{\text{query}}$ to get a dense representation. $\text{Enc}_{\text{doc}}$ encodes all the documents in the retrieval corpus. Once all the documents and the query are encoded, a similarity score is calculated between each document $d$ and the query $q$:

$$\text{sim}(q, d) = \frac{\text{Enc}_{\text{query}}(q)^T \text{Enc}_{\text{doc}}(d)}{\sqrt{h_{\text{dim}}}},$$

where $h_{dim}$ represents the encoding dimension of the retriever encoders. Since the query and doc vectors are not normalized, the inner product can be very large. In order to stabilize the training, we scale the inner product by dividing it by the square root of the hidden dimension.

Once all the similarity scores are calculated, we use softmax to create a probability distribution over all the documents for a given query:

$$p(d|q) = \frac{\exp(\text{sim}(q, d))}{\sum_{d' \in D} \exp(\text{sim}(q, d'))}.$$

Finally, we create the inputs to the reader by appending the representations of each $d$ to $q$, i.e. $[q; d]$. In other words, if we have a retrieval corpus $D$ with $N$ documents, then a single query generates $N$ inputs to the reader — effectively multiplying by $N$ the batch size passed to the reader. However, it is unfeasible to do this for the whole corpus, therefore we only retrieve the top-k documents based on the similarity scores.

**Reader** The reader is a single pre-trained language model taking as input the document $d$ appended to query $q$ ($[q; d]$). During continued pre-training, the reader is optimized for MLM. For each input to the model, we get predictions on what the masked words in the query are — given the context provided by document $d$. Formally, each input generates the following probability for the correct masked words $y$:

$$p(y|d, q) = \prod_{y_i \in M_q} p(y_i|d, q),$$

where $y_i$ is the $i$-th masked word in query $q$ and $M_q$ is the set of all masked words in $q$. However, we want $p(y|q)$. Therefore, for a query $q$, we need to do $k$ forward passes to get all the $p(y|d, q)$. Finally, to obtain $p(y|q)$ we marginalize:

$$p(y|q) = \sum_{d \in D} p(y|d, q)\, p(d|q).$$

**Loss** With $p(y|q)$ we can calculate the loss. During the loss function (cross-entropy) derivation, the error backpropagation is spread between the reader and the retriever. For the reader, this is the same as for any transformer-based model trained on the MLM task except that it averages over the batch size, number of retrieved documents, and the number of masked tokens. For the retriever, it is updated based on whether $p(y|d, q)$ was better or worse than $p(y|q)$. Specifically, if $p(y|d, q)$ is higher than $p(y|q)$, then the similarity score between $q$ and $d$ should increase. This can be seen with the following equation:

$$\nabla_\theta \log p(y|q) = \sum_{d \in D} u(d) p(d|q) \nabla_\theta \text{sim}(d, q)$$

$$u(d) = \left( \frac{p(y|d, q)}{p(y|q)} - 1 \right),$$

where $\theta$ represents the parameters of $\text{Enc}_{\text{doc}}$.[2] The same derivation applies to the parameters of $\text{Enc}_{\text{query}}$.

---

[2]The full derivation can be found in the appendix of Guu et al. (2020), where $z = d$, $x = q$ and $f$ represents the function sim

## 3.2 Null Document

As pointed out in Guu et al. (2020), not all masked words need world knowledge to be predicted correctly. Therefore, we also add a null document appended to the query $q$. There are two ways to encode the null document. The first, and most obvious, is to pass the empty string to $\text{Enc}_{\text{doc}}$ and use the resulting encoding as the null document. However, we use a parameter tensor initialized with all zeros instead. This saves us one forward and backward pass of $\text{Enc}_{\text{doc}}$ per step, without affecting performance.

## 3.3 Corpus

We use a snapshot of the Norwegian Wikipedia from October 2022 as our corpus, limited to the Bokmål written standard. We pre-process the articles into chunks of token length 128, padding the last chunk of each article so that no chunk contains text from two different sources. After processing, the corpus consists of 735 000 documents, with an average number of words per chunk being about 100 ($\mu = 102, \sigma = 25$). We use this corpus for sampling sentences to mask for MLM and as a retrieval corpus during continued pre-training.

## 3.4 Search index

As described in our architecture, we use the documents $d$ in the retrieval corpus $D$ to improve the model's predictions. Ideally, we would use all the documents of the retrieval corpus to make the prediction. Then, the model would assign close to zero probabilities to most documents, while simultaneously having access to all documents, and therefore identifying the most relevant ones. However, this is not feasible, as it would require a very high amount of resources (which would keep increasing as we increase our retrieval corpus) and be unreasonably time-consuming. Therefore, we instead only retrieve the top-k documents in terms of similarity score. To be able to efficiently retrieve and find these documents, we use a search index. We build this index using the encoding of the documents produced by $\text{Enc}_{\text{doc}}$. Since we update $\text{Enc}_{\text{doc}}$ at every backward pass, it follows that we should re-index the documents at each backward pass. However, this is too time-consuming and we therefore only re-index each $s$ steps. We want to note here that there has been recent work on how to more efficiently retrieve from such an index (Alon et al., 2022; He et al., 2021). Since we use the same corpus for both MLM training and retrieval, the first retrieved document is often the same document from which the query comes from, as this will naturally have a high similarity score. To avoid directly giving our model the answer with the unmasked token in it, we make sure to remove this document.

## 3.5 Inverse cloze task

We warm up the encoders for both the query and the document with the ICT task from Lee et al. (2019) on $68k$ Wikipedia article introductions limited to 128 tokens, from a snapshot from October 2020. For each pass, the model must predict the relevant pseudo-document for a pseudo-question from a set of distractors. The question is a random sentence and the document is the text surrounding it, the distractors are sampled from the same batch.

## 3.6 Span Masking

For the MLM task, we combine both salient masking (Guu et al., 2020), where only named entities and dates that require world knowledge are masked, and random masking. We identify entities using an off-the-shelf named entity recognizer and dates with a simple parsing algorithm.[3] We use 15% salient masking, making sure to mask at least one salient span for each sample, and 3.75% random span masking (which is 25% of 15%). By doing this, we encourage the network to learn to retrieve spans requiring world knowledge while ensuring that the model is still able to model linguistic features.

## 4 Experiments

We evaluate BRENT on a wide range of Norwegian NLP tasks. We do this both without retrieval using the extracted reader, and with retrieval turned on using the full model. By doing this, we highlight both the improved capacity of the reader to use context and show how retrieval in general affects performance on NLP tasks other than QA. This section describes the specific datasets and models we use during experimentation.

### 4.1 Models

**NorBERT2**  A baseline Norwegian LM, originating from Kutuzov et al. (2021).

**NorBERT2$_{50k}$**  A NorBERT2 model trained for 50k additional steps on Wikipedia using MLM

---

[3]spaCy: https://spacy.io/

as described in Section 3.6, with a batch size of 1024. We show the performance of this model in order to get a more fair comparison, showcasing the improvements gained from the actual retrieval-augmented pre-training as compared to just doing regular pre-training for 50k more steps on the same corpora.

**BRENT** The entire model with retrieval turned on during fine-tuning. This is akin to a Norwegian version of REALM (Guu et al., 2020), but with our modifications. When subscripted, this indicates the source of the retrieval corpus, which could be either from Wikipedia or a task-specific dataset.

**BRENT$_{reader}$** The reader model extracted after continued pre-training, used without any retrieval during fine-tuning on the downstream tasks.

## 4.2 Datasets

**NorQuAD** A Norwegian question answering dataset for machine reading comprehension (Ivanova et al., 2023) based on the SQuAD format (Rajpurkar et al., 2016). For a given question, the model must predict the correct span in a provided passage that answers the question. NorQuAD includes three domain splits: one sourced from the Norwegian Wikipedia ($N = 2351$), one from Norwegian news articles ($N = 2398$), and one split that combines both of them ($N = 4749$). We use an $80 - 10 - 10$ split on all three domains for training, validation, and testing.

**NoReC$_{fine}$** A fine-grained sentiment analysis dataset for Norwegian (Øvrelid et al., 2020). The texts are a subset of the NoReC dataset (Velldal et al., 2018), a multi-domain dataset of full-text professional reviews published in Norwegian on-line news sources. Each sentence in NoReC$_{fine}$ is annotated for sentiment holders, targets, polar expressions, expression polarities, and polar intensities. A version for targeted sentiment analysis (TSA) is released on GitHub where only the sentiment targets are labeled.[4]

**NoReC$_{sent}$** A sentence-level sentiment analysis dataset for Norwegian derived from NoReC$_{fine}$ (Øvrelid et al., 2020; Kutuzov et al., 2021). This dataset is generated by aggregating the entity sentiments in each sentence. The sentences are then labeled as either positive, negative, or neutral. We

use the version only containing positive and negative sentiments. Both versions of the dataset (with and without neutral sentiment sentences) are available on GitHub.[5]

**Morpho-syntactic tasks** This group of tasks is based on annotations from the Norwegian Dependency Treebank (Solberg et al., 2014), which were converted to the Universal Dependencies (UD) format by Øvrelid and Hohle (2016) and later enriched with named-entity types by Jørgensen et al. (2020). The resulting dataset is called NorNE and we use its latest version.[6] The source of NorNE is mostly news texts, but also government reports, parliament transcripts, and blogs. We evaluate the models on all available UD tasks for Norwegian Bokmål (UPOS and UFeats tagging, lemmatization, and dependency parsing; Nivre et al., 2016),[7] as well as on named entity recognition (NER).[8]

## 4.3 Implementation details

Since running these models is resource intensive, we do not do a hyperparameter search. Instead, we base our hyperparameters on previous research where available. The following paragraphs outline the details of our experiments.

**Search Index** We use the FlatIndexIP from the FAISS (Johnson et al., 2019) library to construct our index. This allows us to get the most relevant documents rather than an approximation of the best documents. We can do this since our corpus of documents is relatively small. We retrieve the top-7 documents and append the null document, in essence retrieving 8 documents in total. We re-index every 100 steps.

**ICT** We use NorBERT2 as the initialization for the ICT warmup. For this, we use a learning rate of $1 * 10^{-4}$ and batch size of $128$ for 10 epochs with early stopping on a single NVIDIA A100 GPU. After the warmup, these weights are then used as the starting point for $Enc_{query}$ and $Enc_{doc}$ in the

---

[4] https://github.com/ltgoslo/norec_tsa

[5] https://github.com/ltgoslo/norec_sentence/

[6] https://github.com/ltgoslo/norne

[7] We use the official evaluation script from CoNLL 2018 shared task (Zeman et al., 2018, https://universaldependencies.org/conll18/evaluation.html).

[8] We employ the evaluation method from SemEval 2013 task 9.1 (Segura-Bedmar et al., 2013), re-implemented in https://github.com/davidsbatista/NER-Evaluation.

| Model | Wiki | | News | | All | |
|---|---|---|---|---|---|---|
| | EM | F1 | EM | F1 | EM | F1 |
| Human* | 72.65 | 88.84 | 83.61 | 93.43 | 78.13 | 91.14 |
| NorBERT2 | $57.76^{\pm1.15}$ | $71.89^{\pm0.89}$ | $64.05^{\pm1.27}$ | $76.93^{\pm1.15}$ | $64.64^{\pm1.40}$ | $77.86^{\pm0.65}$ |
| NorBERT2$_{50k}$ | $59.14^{\pm0.55}$ | $73.98^{\pm1.05}$ | $64.89^{\pm1.44}$ | $77.22^{\pm0.57}$ | $63.88^{\pm0.49}$ | $77.05^{\pm0.55}$ |
| BRENT$_{reader}$ | $\mathbf{62.57}^{\pm1.77}$ | $\mathbf{76.45}^{\pm1.40}$ | $\mathbf{68.10}^{\pm2.87}$ | $\mathbf{80.40}^{\pm1.71}$ | $\mathbf{66.56}^{\pm1.36}$ | $\mathbf{80.01}^{\pm1.16}$ |

Table 1: Results on different domains of the NorQuAD dataset. Results are reported as the mean and standard deviation over five random seeds. *Human performance is the mean performance of two annotators as reported in Ivanova et al. (2023))

| Model | UPOS | UFeats | Lemma | LAS | NER |
|---|---|---|---|---|---|
| NorBERT2 | $\mathbf{98.65}^{\pm0.04}$ | $\mathbf{97.58}^{\pm0.06}$ | $\mathbf{98.18}^{\pm0.03}$ | $\mathbf{93.15}^{\pm0.05}$ | $88.13^{\pm0.34}$ |
| NorBERT2$_{50k}$ | $98.64^{\pm0.04}$ | $97.54^{\pm0.04}$ | $98.12^{\pm0.06}$ | $93.10^{\pm0.22}$ | $\mathbf{88.41}^{\pm0.45}$ |
| BRENT$_{reader}$ | $98.62^{\pm0.06}$ | $97.55^{\pm0.02}$ | $98.09^{\pm0.04}$ | $92.96^{\pm0.15}$ | $87.70^{\pm0.49}$ |

Table 2: Results on the morpho-syntactic tasks: accuracy of UPOS and UFeats tagging, the accuracy of lemmatization, the labeled attachment scores of dependency parsing, F1 scores of named entity recognition, where the evaluation requires an exact match on both span and label. Results are reported as the mean and standard deviation over five random seeds.

retriever, while the reader uses NorBERT2 without any warmup.

**Pre-training** We then train BRENT for 50k steps with a batch size of 1024 divided over 128 AMD MI250X GPUs,[9] a learning rate of $2 * 10^{-5}$, using the AdamW optimizer, and a Cosine scheduler with a warmup, on the chunked Wikipedia corpus. A full description of the model and the hyperparameters can be found in Appendix A.2.

**Fine-tuning** We run all experiments using five different seeds and report the average result and standard deviation. For the fine-tuning of the retrieval-enhanced models, we test both with and without re-indexing, i.e., fine-tuning $\text{Enc}_{doc}$. In both cases, we continue to fine-tune $\text{Enc}_{query}$. When fine-tuning, we use a higher learning rate for the retriever as compared to the reader, since we saw experimentally that this obtained better results. When re-indexing, we do it every 100 steps and at the end of each epoch. Hyperparameters for all evaluation tasks can be found in Appendix A.3. We fine-tune all models on a single GPU.

## 4.4 Results

### 4.4.1 Extractive QA

Table 1 shows the exact match (EM) and token-level F1 scores of different approaches on the

---

[9]These resources were made available to us through the EuroHPC JU project: https://www.lumi-supercomputer.eu/

| Model | BSA F1 % | TSA F1 % |
|---|---|---|
| NorBERT2 | $\mathbf{85.52}^{\pm0.74}$ | $\mathbf{47.58}^{\pm0.49}$ |
| NorBERT2$_{50k}$ | $84.62^{\pm0.50}$ | $46.70^{\pm0.65}$ |
| BRENT$_{reader}$ | $83.33^{\pm0.47}$ | $46.48^{\pm0.26}$ |
| BRENT$_{Wiki}$ | $84.21^{\pm0.37}$ | $44.06^{\pm0.73}$ |
| BRENT$_{Wiki;nri}$ | $84.18^{\pm0.53}$ | $43.38^{\pm1.45}$ |
| BRENT$_{NoReC}$ | $84.35^{\pm0.41}$ | $43.55^{\pm0.26}$ |
| BRENT$_{NoReC;nri}$ | $83.90^{\pm0.56}$ | $44.22^{\pm0.42}$ |

Table 3: Results of the binary sentiment analysis task (BSA) on the NoReC$_{sent}$ dataset and targeted sentiment analysis (TSA) on the NoReC$_{fine}$ dataset. Evaluation is on the test set and is based on the best model found during training. Results are reported as the mean and standard deviation over five random seeds. nri stands for no re-indexing. The NoReC subscript represents the training dataset being used as the retrieval corpus.

NorQuAD dataset. BRENT$_{reader}$ outperforms all other approaches on the three domain splits, especially with respect to the EM metric, which we explain by the salient masking technique. Although BRENT$_{reader}$ was only trained on Wikipedia, the improvement in performance is significant also for questions in the news category. Naturally, NorBERT2$_{50k}$ also improves a bit compared to NorBERT2 on the Wikipedia split, but not by the same margin, and not at all on the news category. This indicates that BRENT$_{reader}$ actually learns to use context better, that this generalizes beyond

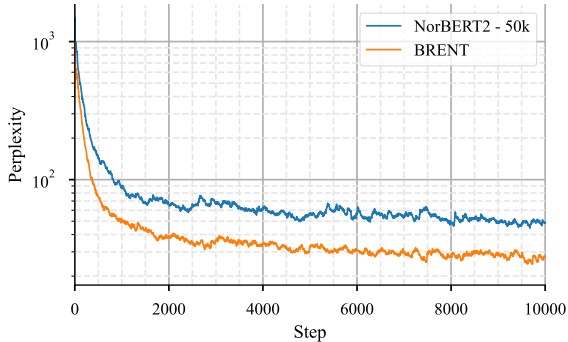

Figure 2: Training perplexity during the first $10\,000$ steps. The values are smoothed with an exponential moving average, using $\alpha = 0.99$.

the style of Wikipedia, and that this could not be achieved by simply training the same underlying LM for 50k additional steps on the same corpus with the same MLM setup.

### 4.4.2 Sentiment analysis

As for sentiment analysis, Table 3 shows that $\text{BRENT}_{\text{reader}}$ performs worse compared to the baseline of NorBERT2 on the binary sequence classification task, indicating that the continued pre-training with retrieval does not actually help for this task, but rather impedes performance. This is also the case for the $\text{NorBERT2}_{50k}$ model, albeit with a smaller impediment to performance, suggesting that it might be the continued training on Wikipedia reducing the performance of the models on this task. When retrieval is used, as can be seen in the bottom half of Table 3, the performance is better, but still short of the baseline. For TSA, the reader performs on par with the baselines but turning retrieval on substantially decreases performance.

When retrieving from a corpus during fine-tuning, our model retrieves reviews that are related with respect to inner product similarity, not necessarily sentiment. If classifying a negative review of a TV, our model could end up retrieving another review about some other electronic apparatus — which might be positive. This is clearly not helpful for the task at hand. In order to teach the retrievers to retrieve based on sentiment, we would need a bigger dataset to fine-tune on. Despite this, manual inspection shows that the retrieved contexts are sometimes very relevant for the query when the retrieval corpus is NoReC. When the retrieval corpus is Wikipedia, however, the contexts are of low relevance. Examples of queries and retrieved contexts for $\text{BRENT}_{\text{Wiki}}$ and $\text{BRENT}_{\text{NoReC}}$ on binary sentiment analysis (BSA) can be found in Appendix A.1.1 and Appendix A.1.2.

We also note that not re-computing the search index decreases performance. However, as performance is relatively similar, it might not be worth it as re-indexing is a lot more resource-demanding. With Wikipedia as the retrieval corpus on our computing setup, TSA fine-tuning takes about 7 hours with re-indexing, compared to 2.5 hours without.

### 4.4.3 Morpho-syntactic

Table 2 shows the results of the reader compared to the baselines on a series of Norwegian token-level labeling tasks. $\text{BRENT}_{\text{reader}}$ performs on par with the baseline models, which strengthens our hypothesis that the continued pre-training with retrieval does not impede the model's ability to perform morpho-syntactic tasks while simultaneously increasing performance on extractive QA. This claim is further supported by the fact that the same happens with $\text{NorBERT2}_{50k}$, which indicates that adding the retrieval is no worse than just continuing to do MLM over the same corpus for additional steps.

### 4.5 Analysis of the pre-training

Figure 2 shows the perplexity values of BRENT and $\text{NorBERT2}_{50k}$ during the first 10k steps of continued pre-training on the Wikipedia corpus. After the initial convergence phase, $\text{NorBERT2}_{50k}$ establishes itself on values around 40, while BRENT sits at around 20. As we do mainly salient masking, perplexity is a proxy for how well the models predict the correct named entities and dates. The difference between the two shows how retrieval is helpful for predicting masked entities.

## 5 Ablations

As with other retrieval-augmented LMs, BRENT is a pipeline model — consisting of multiple parts that interact according to a series of design choices that impact the outcome. Due to the computational cost of pre-training, it is not feasible to quantitatively determine the effect of all these choices, resulting in a poor understanding of some aspects of these models. To mitigate this, we study the effect of some of these choices with respect to the overall loss during pre-training with a series of ablations. We do this for a reduced number of steps, but with

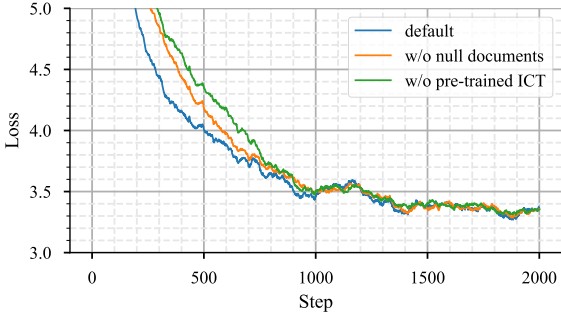
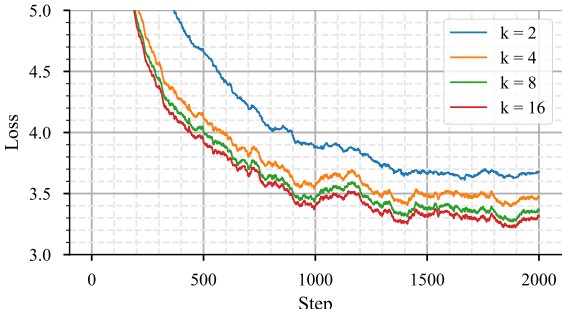

Figure 3: Loss curves of having no null document or no ICT warmup compared to the tested model. The values of all three runs are smoothed with an exponential moving average, using $\alpha = 0.99$.

Figure 4: Loss curves of the first $2\,000$ training steps with varying number of retrieved documents $k$. The values are smoothed with an exponential moving average, using $\alpha = 0.99$.

the same retrieval corpus and with the same GPU setup as described in Section 3.3 and Section 4.3.

## 5.1 ICT

Figure 3 shows the effect of the ICT warmup task with respect to the loss for 2000 steps. When ICT is turned off, $\text{Enc}_{\text{doc}}$ and $\text{Enc}_{\text{query}}$ are initialized with the same weights as the reader. As can be seen from the figure, the loss converges slower when the ICT task is not used, but it is quickly matching the setting when it is used. Guu et al. (2020) claims that without ICT one would encounter a cold-start problem where the retrieved documents will likely be unrelated to the query at the beginning of training, causing a cycle where the encoders do not receive meaningful gradients. We find that this is not the case and that the effect of ICT warmup is minimal.

## 5.2 The effect of the null document

As mentioned in Section 3.2, we use a parameter tensor initialized with all zeros for representing the null document. This is optimized jointly with the rest of the weights. Figure 3 shows how the model behaves when the null document is removed, which is done by making the probability of the null document zero, as compared to the test model which has it included. Contrary to Guu et al. (2020), we find that the effect of the null document is questionable. It makes sense to have a "sink" to use when no retrieval is necessary, but we do not find the null document to fulfill this need.

## 5.3 Varying the number of documents to retrieve

Figure 4 shows how the number of retrieved documents influences training with respect to loss. For the first $2\,000$ training steps, $k = 16$ converges a bit quicker than the $k = 8$. However, we see that the result is minimal after that point, which is also the conclusion in Guu et al. (2020). Given that it is more computationally expensive to train with a higher $k$ and that the gain of going from 8 to 16 is negligible, we keep $k$ at 8.

## 6 Conclusion

We develop the first Norwegian retrieval augmented language model, BRENT, based on the REALM method proposed by Guu et al. (2020). The model uses an encoder-reader architecture, and we train it on a relatively small corpus consisting of 735k Wikipedia documents. In addition to the model itself, our contribution has been to demonstrate how the use of continued pre-training with retrieval benefits the context utilization of the reader, which we extract from the pipeline. The reader performs better than comparable baselines on the extractive QA task without losing performance on morpho-syntactic tasks. We also evaluate our full retriever model on sentiment analysis with two different corpora as the retrieval corpus, but here we observe a decrease in performance overall. Contrary to some previous work, our ablation studies find that the effect of having a null document and using ICT as a warmup task is minimal.

# 7 Future work

A future direction for our work is to study in greater detail how retrieval influences the language modeling task. In particular, we would like to train a retrieval model from scratch. Another direction, which has also been pointed out in related work, is to experiment with cross-lingual retrieval, especially in the case where the retrieval corpus is from a high-resource language. This would be useful in scenarios where a large knowledge source like English Wikipedia could be used to augment a lower resource language, like Norwegian, which does not have such an extensive source available.

## Acknowledgements

Parts of the work documented in this publication have been carried out within the NorwAI Centre for Research-based Innovation, funded by the Research Council of Norway (RCN), with grant number 309834.

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

# A   Appendix

## A.1   BSA retrieval examples

### A.1.1   Wikipedia examples

Examples of retrieved contexts from the BRENT$_{\text{Wiki}}$ model fine-tuned on the BSA task with Wikipedia as the retrieval corpus.

- Query: *Men så kommer de skjærende lydene* 'But then the squeaky sounds appear'

  - Retrieved context: *Øynene* 'The eyes'.

- Query: *Broen går seg vill i sitt eget ønske om å vært artsy* '"The Bridge" is lost in its own wish to be "artsy"'

  - Retrieved context: *Sverige* 'Sweden'

### A.1.2   NoReC examples

Examples of retrieved contexts from the BRENT$_{\text{NoReC}}$ model fine-tuned on the BSA task with the $NoReC$ dataset as the retrieval corpus.

- Query: *Men så kommer de skjærende lydene* 'But then the squeaky sounds appear'

  - Retrieved context: *Allerede under første låt får vi slengt alle klisjeene i trynet.* 'Already during the first song we are hit in the face with all the clichés'.

- Query: *Begeistringen var uvanlig stor og applausen deretter da det 70 minutters lange verket var fullført* 'The enthusiasm was unusually great and so was the applause that followed when the 70-minute long piece was over'.

  - Retreived context: *En helt utrolig konsertopplevelse* 'A wonderful concert experience'.

- Query: *Å være eksperimentell er ikke positivt i seg selv; de mange sjangrene og retningene i musikken gjør helehetsinntrykket rotete og meningsløst* 'Being experimental is not positive in and of itself; the many genres and directions makes the music seem messy and meaningless'.

  - Retrieved context: *Automatisk to-soners klimaanlegg* 'Automatic two-zone aircondition'.

## A.2   Model

The hyperparameters used for the continued pre-training can be found in Table 4.

## A.3   Hyperparameters

### A.3.1   NorQuAD

For comparison, we use the same set of hyperparameters as in Ivanova et al. (2023), visible in Table 5.

### A.3.2   Sequence labeling

For the task of targeted sentiment analysis, we fine-tune and report the average test results over five runs, from the epoch providing the best results on the development set. Hyperparameters can be found in Table 6.

## A.4   Binary Sentiment Analysis

For the task of binary sentiment analysis, we fine-tune for three epochs and select the best model based on the development set's f1 score. We average our test results over five runs. Hyperparameters can be found in table Table 7.

## A.5   Morpho-syntactic

For morpho-syntactic tasks, we fine-tune for 10 epochs and select the best model based on the average performance on the development split. We average our test results over five runs. Hyperparameters can be found in Table 8.

| Hyperparameter | Value |
| --- | --- |
| Number of parameters | 125M |
| Number of attention heads | 12 |
| Number of layers | 12 |
| Hidden dimension ($h_{dim}$) | 768 |
| Activation function | GeLU |
| Vocabulary size | 50104 |
| Seed | 42 |
| Dropout | 0.1 |
| lr | $2 * 10^{-5}$ |
| Weight decay | 0.1 |
| Batch size | 1024 |
| $k$ | 8 |
| re-indexing frequency | 100 steps |
| Steps | 50k |
| Scheduler | Cosine with warmup |
| Warmup | 800 steps |
| Final lr | $2 * 10^{-6}$ |
| Optimizer | AdamW |
| Index | FlatIndexIP |

Table 4: Hyperparameters for the continued pre-training of both BRENT and NorBERT2. $k$ represents the number of documents retrieved including the null document.

| Hyperparameter | Value |
| --- | --- |
| Batch size | 16 |
| Epochs | 3 |
| lr | $5 * 10^{-5}$ |
| Scheduler | Linear |
| Optimizer | AdamW |
| Seeds | $[42, 437, 4088, 3092, 9720]$ |

Table 5: Hyperparemeters for fine-tuning on the NorQuAD dataset

| Hyperparameter | Value |
| --- | --- |
| Batch size | 32 |
| Epochs | 8 |
| $lr_{reader}$ | $5 * 10^{-5}$ |
| $lr_{retriever}$ | $1.5 * 10^{-4}$ |
| $k$ | 4 |
| re-indexing frequency | 100 steps + end of epoch |
| Scheduler | Linear |
| Optimizer | AdamW |
| Seeds | $[101, 202, 303, 404, 505]$ |

Table 6: Hyperparemeters for fine-tuning on the NoREC$_{fine}$ dataset. $k$ represents the number of documents retrieved including the null document. The $lr_{reader}$ is for both the retrieval and non-retrieval models.

| Hyperparameter | Value |
| --- | --- |
| Batch size | 32 |
| Epochs | 3 |
| $lr_{reader}$ | $1 * 10^{-5}$ |
| $lr_{retriever}$ | $3 * 10^{-5}$ |
| $k$ | 4 |
| re-indexing frequency | 100 steps + end of epoch |
| Scheduler | Cosine |
| Optimizer | AdamW |
| Seeds | $[42, 456, 78463, 27485, 34586]$ |

Table 7: Hyperparemeters for fine-tuning on the NoREC$_{sent}$ dataset. $k$ represents the number of documents retrieved including the null document. The $lr_{reader}$ is for both the retrieval and non-retrieval models.

| Hyperparameter | Value |
| --- | --- |
| Batch size | 32 |
| Epochs | 10 |
| $LR_{reader}$ | $1 * 10^{-4}$ |
| $LR_{heads}$ | $1 * 10^{-3}$ |
| Scheduler | Cosine |
| Optimizer | AdamW |
| Seeds | $[1234, 2345, 3456, 4567, 5678]$ |

Table 8: Hyperparemeters for fine-tuning on the morpho-syntactic tasks. $k$ represents the number of documents retrieved including the null document. The learning rate is different for the fine-tuned language model and for the classification heads.