# OpenReview forum: "BRENT: Bidirectional Retrieval Enhanced Norwegian Transformer"
_NoDaLiDa/2023/Conference — NoDaLiDa 2023_

### Official Review · Reviewer_WdED · 2023-03-08
**Unclear paper describing a Norwegian adaptation of the Retrieval-Augmented Language Model**

**Rating:** 4
**Confidence:** 4

**Review:**

The paper describes the adaptation of the REALM model of Guu et al. 2020 to Norwegian.

The key idea of the REALM model is to capture knowledge using a latent "knowledge retriever": for a given query (e.g. for openQA task), the model can retrieve pertaining documents from a document database (e.g. wikipedia) and use these documents as additional context to the query.

Guu et al. provide an analysis of the update step of the retriever, and show that if the probability of masked words in the query q increases when knowing a retrieval document d, then the loss encourages the encoding of d to be similar to that of q.

Instead of OpenQA (like in Guu et al.) the authors test their model on extractive QA in Norwegian, and do report a better performance, showing that the model actually learns to better use context.
An originality of the work is to test the retrieval-enhanced model on various other classic downstream tasks. Performance is degraded for sentiment analysis, but is not degraded on a series of morpho-syntactic tasks.

Overall, I think that the paper describes a model that is valuable for the NLP community, although it is "simply" an adaptation of an existing model.

I think though that the writing and structure of the paper is problematic. The authors chose to mix the description of their system to that of Guu et al., and in the end, it is quite difficult to separate innovations from existing features in REALM.

More importantly, the description of the REALM system is quite unclear, reading Guu et al. is totally required in order to understand the current paper. Even if it could be an option to require that the readers be familiar to REALM, I still find that the description of REALM in the current submission could be substantially improved.

So it seems to me that splitting section 3 by first describing REALM and then describe the modifications performed in the proposed system could be a solution to both flaws.


More detailed questions and comments:


- the abstract and conclusion should mention that the retrieval-enhanced model degrades performance on sentiment analysis. It is misleading not to state it.


Figure 1: although linguistic diversity is obviously quite important, I think not providing the translations makes figure 1 a lot less useful. If place is lacking, then I would prefer a litteral English translation.
Moreover, I find that changing the terminology of Guu et al. is misleading (Neural knowledge retriever => "retriever"; and "knowledge-augmented encoder" => "reader"). I must say Figure 1 of Guu et al. seems a lot clearer to me.

- line 249: maybe explain that the training signal for the Enc_doc is in the reader phase.

- line 276: "our reader": clarify what is specific to BRENT, and what is simply copied from Guu et al. Here it seems the reader is exactly the "knowledge-augmented encoder" of Guu et al.

- lines 305-309: I found this part quite unclear, even though it corresponds to a crucial claim of Guu et al.. It should be said explicitely that it is an analysis of the gradient aiming at understanding how the gradient will modify sim(d,q) : Guu et al. state very clearly that sim(d,q) will be increased iff p(y|d,q) > p(y|q).

- line 328: "we also add a null document": again, it is unclear if this is done in Guu et al. 2020 or not.


**Paper Type:**

Long paper

---

### Official Review · Reviewer_oR46 · 2023-03-10
**Review of BRENT: Bidirectional Retrieval Enhanced Norwegian Transformer**

**Rating:** 7
**Confidence:** 4

**Review:**

# Summary

This paper presents BRENT, a retrieval-augmented language model trained and tested on Norwegian. BRENT is a reimplementation of REALM in a more limited scenario. The results show that having a retriever would benefit the language model in extractive question answering while not hurting the performance on named entity recognition and yielding a slightly worse performance on sentiment analysis. The paper involves numerous ablation studies, e.g., experimenting with the language model with the retrieval mechanism stripped from it, the effect of a warmup method in training, and how the number of retrieved neighbours would affect the loss.

# Strengths

The paper is well-written. It is clear how the models are implemented, trained and tested in most cases. The paper also has extensive appendices that include details from more experiments. Additionally, the accompanying figures facilitate an understanding of the methodology and the experiments.

The paper has clear contributions. It is easy to understand where the proposed method stands compared to the previous work.

Several ablation studies shed light on how different components work in the methodology.

# Weaknesses

It is not surprising that the model works for Norwegian. After all, there is no fundamental difference between Norwegian and English about the task of language modelling and retrieval augmentation.

It is unclear why the results in Section 5.1 are how they are. It is worth investigating why ICT warmup does not work as the cited papers. One reason could be that in BRENT the pretraining objective and the nature of the pretraining and retrieval data are different compared to REALM. In the paper at hand, the pretraining data is the same as the ICT data, i.e., Wikipedia, which could be one reason why ICT is redundant in this scenario. Another reason might be that there is a lot of repetition in Wikipedia, which makes for better neighbours. It would then be no surprise that these methods do not work.

# Comments

1. What do UPOS and UFeats stand for?
2. What do you mean by “closed question answering”? Is it closed-domain QA or something different?
3. You state that the model performs on par with the baselines on other tasks. What about sentiment analysis? Don’t you show that it is not the case for this task?
4. What is the role of root of $h_{\mathrm{dim}}$ in the denominator of $\mathrm{sim}(q,d)$? It is not in REALM, and you do not explain it here.
5. It is unclear how some hyperparameters are chosen, such as the learning rates.
6. The third contribution would be better understood if it were more closely related to the main storyline of the paper. Clarifying its connection to the rest of the content would improve the paper's coherence.
7. Line 355: Why is it “ideal” to have all the documents as neighbours? Is that not confusing for the model when almost all the neighbours are irrelevant?


**Paper Type:**

Long paper

---

### Official Review · Reviewer_ddh1 · 2023-03-10
**Interesting model, good presentation**

**Rating:** 7
**Confidence:** 3

**Review:**

The paper presents a retrieval-augmented language model for Norwegian, which performs well compared with relevant baselines. The experiments appear thorough and are well presented. The paper is relatively well written. I believe it should be accepted for publication.

Miscellaneous comments:

- I assume $1e-4$ on line 517 is meant to be scientific notation for $0.0001$. I am of the opinion that it is more correct, and better, to write it as $10^{-4}$. The authors nicely illustrate the confusion that writing "1e-4" instead can give rise to, by giving the learning rates in the tables as $2e^{-5}$ etc. Unless this is actually meant to be $2 \cdot 2.71^{-5}=0.014$ and not $0.00002$, it really needs to be corrected.

- I would advise against highlighting the references with colored squares, as this impedes readability.

- Abbreviations should generally be avoided in abstracts. The abbreviation «RLMS» is non-standard, somewhat strange (should it be «RLMs»?) and not used elsewhere in the article. «QA», «PoS» and «NER» are standard, but should ideally be written out. (However, I would leave «REALM» as it stands.)

- 026: «which we call the _reader_» It is unclear here what «which» refers to. Rephrase.

- 140: «For the first time, to our knowledge, we study». Bad English, rephrase.

- 148: This bullet point also needs rephrasing.

- 177: transformer model (not «Transformers model»)

- 216: write out the abbreviation DPR

- 256: «dim» should be in text mode

- 427: 3.6, (missing comma)

- 570: i.e., (missing comma)

- 690: Why is «NoReC» italicized?

- 1186: morpho

- 1231: the placement of these lines is clumsy. Maybe it would be better to move Table 4?


**Paper Type:**

Long paper

---

### Decision · Program_Chairs · 2023-03-17

Accept